# Establishing a taxonomy of potential hazards associated with communicating medical science in the age of disinformation

David Robert Grimes ,[1,2] Laura J Brennan,[3] Robert O'Connor[4]

Dr Laura J Brennan deceased on 20 March 2019.

[1]School of Physical Sciences, Dublin City University, Dublin, Ireland
[2]Department of Oncology, University of Oxford, Oxford, Oxfordshire, UK
[3]Not applicable (patient advocate, deceased), Ennis, Munster, Ireland
[4]Irish Cancer Society, Dublin, Ireland

**Correspondence to**
Dr David Robert Grimes; davidrobert.grimes@dcu.ie

## ABSTRACT

**Objectives** Disinformation on medical matters has become an increasing public health concern. Public engagement by scientists, clinicians and patient advocates can contribute towards public understanding of medicine. However, depth of feeling on many issues (notably vaccination and cancer) can lead to adverse reactions for those communicating medical science, including vexatious interactions and targeted campaigns. Our objective in this work is to establish a taxonomy of common negative experiences encountered by those communicating medical science, and suggest guidelines so that they may be circumvented.

**Design** We establish a taxonomy of the common negative experiences reported by those communicating medical science, informed by surveying medical science communicators with public platforms.

**Participants** 142 prominent medical science communicators (defined as having >1000 Twitter followers and experience communicating medical science on social and traditional media platforms) were invited to take part in a survey, with 101 responses.

**Results** 101 responses were analysed. Most participants experienced abusive behaviour (91.9%), including persistent harassment (69.3%) and physical violence and intimidation (5.9%). A substantial number (38.6%) received vexatious complaints to their employers, professional bodies or legal intimidation. The majority (62.4%) reported negative mental health sequelae due to public outreach, including depression, anxiety and stress. A significant minority (19.8%) were obligated to seek police advice or legal counsel due to actions associated with their outreach work. While the majority targeted with vexatious complaints felt supported by their employer/professional body, 32.4% reported neutral, poor or non-existent support.

**Conclusions** Those engaging in public outreach of medical science are vulnerable to negative repercussions, and we suggest guidelines for professional bodies and organisations to remedy some of these impacts on front-line members.

## Strengths and limitations of this study

► Individuals prominently involved in the communication of medical science across different media were surveyed to ascertain their experiences in public outreach.
► Participants were from around the world, but predominantly communicated in the English language.
► Self-selection bias in this survey is unavoidable, and findings cannot be taken as generalisable.
► Accordingly, survey results should only be taken as indicative of the scope of the issue at this juncture.
► Much further research is needed to ascertain how the medical community can best act to counter the rise of medical disinformation while protecting practitioners.

distrusted. Disinformation undermining health science and evidence-based medicine has increased markedly in the era of social media, and dangerous misconceptions abound, from perceived cancer risks and ostensible cures[1] to dangerous falsehoods about vaccination.[2] Improving public awareness and understanding of science and medicine is imperative if we are to maintain continued progress in research endeavours, and scientists, physicians and science communicators have a crucial role to play in shaping public perceptions. Medical science is largely publicly funded, and direct communication of research with the wider public can be extraordinarily beneficial on a societal level. Accordingly, public engagement has become a prerequisite for many funding bodies. Informed engagement by patient advocates and media figures too can have marked impact on public understanding of medicine, empowering the public with facts with which to make important health decisions.

Improving public understanding of medical science is vital, as there many scenarios where public perception (or a vocal subset of that)

## INTRODUCTION

Despite being fundamental to societal well-being, many aspects of medical science remain poorly understood and frequently

is starkly at odds with scientific consensus. Frequently, medical science contradicts a narrative strongly held by particular groups within the wider public. For our purposes, we define a 'narrative' as a world view or mindset shared by a given subgroup, which unifies that grouping. Narratives are often articles of faith, empowerment or comfort, frequently unsupported by available evidence or at odds with scientific consensus. For clarity, we concentrate herein on situations where there is no reputable evidence for a narrative, or where overwhelming scientific consensus is firmly against that viewpoint.

Misguided narratives can be supremely damaging, and the antivaccine movement is perhaps the most obvious example of this. Despite the life-saving efficacy of vaccination, opposition has existed since the time of Jenner.[3] The rise of social media has seen significant propagation of antivaccine narratives,[4–6] driving uptake rates down and causing serious harm worldwide.[7 8] In 2018, Europe saw the highest number of cases of measles in 20 years, numbering over 82 525 cases with at least 72 deaths—over 15-fold the figures from 2016.[9] Such is the extent of the problem that in 2019 the WHO described vaccine hesitancy as a 'Top ten threat to global health'.[10] Exposure to antivaccine conspiracy theory is a leading factor in parental intention to vaccinate,[8] and evidence to date suggests that the deluge of vaccine disinformation across social media is extremely damaging to public understanding and health.

Other strongly held narratives which clash with the weight of available scientific evidence include the claims propounded by the antifluoride movement,[11 12] the beliefs of the electromagnetic hypersensitivity movement[13] and the narratives of complementary medicine.[14 15] Patients with cancer are especially vulnerable to misinformation, and frequently targeted by charlatans and the misguided.[16] Consequences of this can be severe, with patients sometimes delaying or refusing conventional treatment. The net result of this is diminished survival statistics for those who engage with cancer pseudoscience, due to delayed treatment and sometimes rejection of conventional medicine. In some instances, subscribing to unproven or disproven modalities could approximately half survival time.[17]

While health falsehoods have always existed, the social media age has created new avenues for misinformation (misinformed advice) and disinformation (deliberate falsehoods) to propagate,[16] rapidly bringing discredited ideas and dangerous pseudoscience to vast new audiences. Scaremongering stories from dubious outlets propagate more readily than reliable fact-based information from reputable sources.[18 19] In 2016, over half of all cancer stories shared on Facebook were medically unsound. Some have harnessed pseudoscience to sell questionable diets, supplements and books, to the detriment of public understanding. Internet health guru Joseph Mercola, for example, made over $7 million in 2010 alone, proffering dubious treatments and advice,[20] including denigration of conventional therapies for cancer. Mercola is far from unique, and such proclamations have huge potential for patient harm.

To overcome this challenge, public outreach by scientists, physicians and evidence-based health advocates must be a crucial element to counter damaging fictions and empower our community with evidence-based information. A physician's recommendation, for example, is central to parental decisions to vaccinate.[21] Addressing patient concerns improves public health, and personal engagement by researchers and physicians can have a positive impact on public perception. Patient advocates and media figures have substantial ability to shift public perception; after Ireland saw human papillomavirus (HPV) vaccine uptake drop from 87% to 51%, an alliance of healthcare professionals, researchers and patient advocates were instrumental in countering the dominant falsehoods, and Ireland has seen a dramatic recovery in vaccine uptake rates.[22] To make inroads against the deluge of dubious health claims to which we are subjected, it is vital that scientists, clinicians and patient groups must be on the vanguard of efforts to counteract misinformation.

Those engaging in public outreach, however, often encounter enmity for publicly advocating scientific evidence. Scientific consensus often runs contrary to deeply held beliefs, leading to certain groups attempting to undermine legitimate public scientific discourse. Motivations for this are multifaceted, often depending on very specific circumstances. Conspiratorial thinking underpins many narratives, and those attempting to communication science are often vilified as 'shills', or agents of a nebulous 'Big Pharma'. The phenomenon of identity protective cognition is also commonly encountered[23] and narrative believers frequently attacking those who cast doubt on their beliefs. Even when handled with sympathy and compassion, professional and patient advocates who challenge misconceptions can become targets for certain individuals and groups. These negative responses can range from verbal abuse to coordinated harassment campaigns, and even violence.

Aside from being deeply unsettling, such responses can cause professional and personal problems for those targeted. With the increasing emphasis on public engagement and ubiquity of the internet, this subject warrants urgent consideration, as there are currently few clear guidelines for researchers, clinicians or patient advocates engaging in outreach work. Nor indeed is there a unified understanding of how adverse effects can manifest, and institutions and professional bodies are typically ill equipped or muted in their support. This leaves those in the public eye or studying contentious topics vulnerable to vexatious complaints and even physical harm. Without awareness of this reality, a less than ideal situation where professional bodies can potentially be weaponised against researchers can too easily emerge. Accordingly, the aim of this work is to ascertain the typical experiences of those communicating medical science and identify how negative impacts might be counteracted.

## METHODS

### Sample recruitment and selection criteria

The main recruitment fora for this study were online discussion groups for physicians, scientists and patient advocates communicating aspects of medical science to the general public across social and traditional media. At the height of the Irish HPV vaccine confidence crisis,[22] several physicians and scientists (based in Ireland and the UK) in these groups sought advice for negative experiences, including threats and malicious complaints to their employers and professional bodies, all of which were eventually dismissed. Group members across Europe echoed similar experiences in communicating vaccination science and in other health issues, and almost identical adverse reports came from colleagues across the Americas and Africa. Informal interviews were conducted on foot of this in these fora to identify common issues, as to the authors' knowledge there is no existent literature on the topic.

Based on these interviews and related fora discussions, a survey was created, including free-form sections where subjects were free to expand on their own experiences. The wording of this survey is included in the online supplementary material. The participant selection criteria were specifically for those communicating medical science both on social media (defined as having over 1000 followers on Twitter) and in mainstream channels (defined as invited appearances on public television, radio and/or in the form of newspaper articles and invited comment). With this participant selection criteria, 142 individuals worldwide (from across Europe, America, Africa and Asia) working predominantly in the English language were identified and invited to partake, of whom 101 (71.1%) responded. This survey was undertaken for indicative purposes and was collected from a non-randomised group with no expectation of transferable findings. Accordingly, the Health Research Authority decision tool (online at http://www.hra-decisiontools.org.uk/research/) indicated specific ethical approval was not required, with the research governance body of Queen's University Belfast (the lead author's primary affiliation at the time) confirming ethical approval for the survey was not required. In all cases, informed consent was sought and obtained prior to subjects partaking, with all data appropriately anonymised. Subject details are given in table 1.

### Patient and public involvement

As patient advocates play a substantial role in combatting misinformation on medical issues, several who met the inclusion criteria were invited to take part, with 15.8% of respondents being patient advocates. One coauthor of this work (LJB) was a prominent patient advocate.

## RESULTS

One hundred and forty-two individuals were approached to undertake the survey, with 101 responding (response rate: 71.1%). In addition to survey questions on known

| Table 1 | Participant details |
| --- | --- |
| Inclusion and completion | Invited to take part (n=142) |
| | Total completed (n=101) |
| Gender | Female (n=55) |
| | Male (n=44) |
| | Non-binary/undisclosed (n=2) |
| Affiliation | University/medical centre (n=52) |
| | Unaffiliated (n=26) |
| | Media organisation (n=20) |
| | Charity (n=11) |
| | Political organisation (n=4) |
| Profession | Medical professional (n=23) |
| | Scientist/researcher (n=20) |
| | Science communications (n=16) |
| | Patient advocate (n=16) |
| | Health policy (n=5) |
| Years active | Average: 10.7 years (range 2–30 years) |

problems, participants were also invited to expand on noteworthy negative situations they may have encountered while engaging in health outreach. This section was entirely optional, and 53 participants (52.5% of subjects) opted to share their experiences. Example responses are shown below; please note that some responses have been edited or partially redacted to exclude potentially identifying information and preserve anonymity.

> Accusations—including by one Senator—that [we are] uncaring, dismissive, neglectful, arrogant, or paid by pharma companies when advocating for vaccines. (Misrepresentation)

> I find my expertise is questioned—this often seems to be when men find it difficult to accept women with intelligence and qualifications. Sexist insults are a typical go-to response. (Discreditation)

> The worst one that hurt me professionally and personally was that activists gathered my emails using [Freedom of Information Requests] and handed chosen packets of them with a story to different reporters. (Misrepresentation/Discreditation/Dubious Amplification)

> Persistent negative comments on twitter; usually it doesn't last long but it can feel very intense while it's happening! (Intimidation)

> I have been served with a SLAPP lawsuit in order to silence my outreach work. Frequently receive harassing emails, malicious comments made on blog. (Malicious Complaints)

> Social media co-ordinated intimidation, implied threats of legal action (for defamation). Mocking, undermining, condescension and attacks for being an industry shill, although. I am just a patient advocate. Being called a liar, that I never had cancer, that I

deserved cancer due to my attitude, that I have been mutilated by conventional medical treatment, and that I am no longer a woman (having had mastectomy for cancer). That my cancer will return and I deserve that. (Dubious Amplification/Misrepresentation/Discreditation)

I have had anti-vaccine organizations and individuals attempt to prevent my public appearances and have been the subject of numerous online smear campaigns accusing me of being 'a shill for Big Pharma' etc. (Discreditation/Dubious Amplification)

Those who attack me very frequently try to do it by targeting me at my job, sending bogus complaints to my bosses and the university. From my observation, that is the go-to attack, the first thing these groups do. (Malicious Complaints)

I had to contact the police, who visited the person who was harassing me. I also involved social services. We bought a CCTV to monitor our front door after a strange envelope was hand delivered. The person involved has targeted several people before and continues to target individuals who advocate vaccination. (Intimidation)

Abuse and accusations of corruption are the most common adverse reaction I get. Sometimes a particular group petition one's employer and try to create trouble for them. I have been lucky in the past when this happened to have had supportive universities who appreciate my outreach work. I have in the past had slightly unhinged individuals writing rambling, implicitly threatening letters to my office which ultimately required police intervention. (Discreditation/Malicious Complaints/Intimidation)

The worst are gendered insults (being called a cunt, etc.) and rape/death threats. I have had one empty legal threat that was widely publicized. (Intimidation/Malicious Complaints)

Regular threats to sue for defamation. (Malicious Complaints)

Attempts to get me fired, public records act requests for emails, verbal attacks on my children. (Malicious Complaints, Intimidation)

One of the most unpleasant things is that certain people or groupings will use very underhanded tactics to respond to perceived criticism. If they can't refute the science, it isn't uncommon for them to go after you personally, alleging all manner of things to anyone who'll listen; that you're incompetent, or unethical, or perverted. It seems they throw things wildly to see what sticks, but it can be extraordinarily unpleasant to endure. (Dubious Amplification/Discreditation)

My main concern has been obsessed individuals who declare their enmity and seem to be unconstrained by civil norms. (Intimidation)

Homeopathy advocates looked up my LinkedIn profile and called my employer to complain about my comments on the radio. My employer did not support me and I ended up having to stop the activity I had been planning. (Misrepresentation/Malicious Complaints)

Being threatened with physical violence. (Intimidation)

A delusional supporter of [an individual] I wrote about accused me and my lawyer of stalking him and killing his in-laws. He sent accusing emails to the faculty of my school and all the police departments in my state. [They] also accused me of being a terrorist and complained about me to the FBIs Terrorism Joint Task Force. That gave me many nervous, sleepless nights. (Discreditation/Malicious Complaints)

Death threats received, employer unhelpful, sorted myself. (Intimidation)

I haven't experienced many negative encounters because I would say I am only lightly involved in public engagement. However the reason I don't become more heavily involved in this area is fear of this kind of abuse and vexatious complaints to my employer or regulatory body. (Malicious Complaints)

Based on this and survey responses, a non-exhaustive taxonomy was constructed detailing common experiences of those communicating medical science to the public. While non-exhaustive, it forms a useful basis for more systematic investigation. Adversarial experiences in communicating medicine were broadly stratified into five distinct classes, illustrated graphically in figure 1, with these subtypes detailed in table 2.

Participant details are given in table 1. Topics covered, channels of engagement and fora for abusive interactions are depicted in figure 2, informed by survey questions 4,

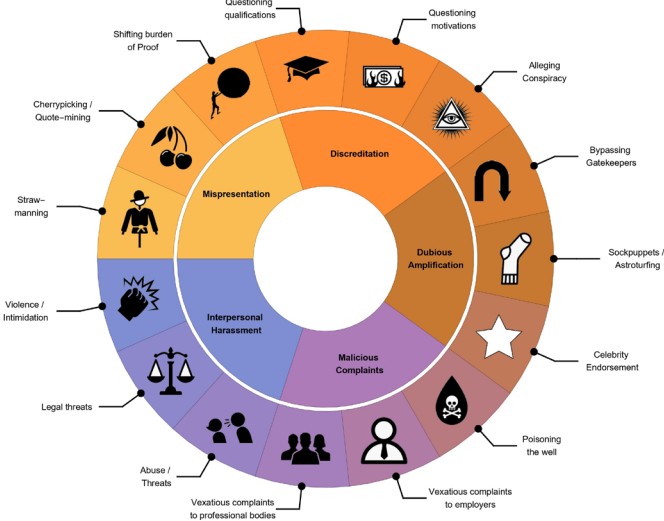

**Figure 1** A non-exhaustive taxonomy of negative experiences encountered by individuals engaging in public communication of health science. Subsections discussed in text.

**Table 2** A non-exhaustive taxonomy of common adversarial tactics

**Misrepresentation of scientific evidence/expertise**

| | | |
|---|---|---|
| Straw-manning | Misrepresenting scientific arguments to make them amenable to ridicule or attack, and to deflect or obscure evidence that undermines a particular narrative. | *'Mercury is toxic, yet scientists put it in vaccines!'*—This statement belies importance of dose and ignores the fact there is no evidence for harm from thimerosal in vaccines. |
| Cherry-picking/quote mining | Selective, manipulative filtering of scientific evidence or expert statements, taken out of context to undermine evidence base or promote a narrative. | *'THC kills cancer, but doctors don't want you to know about cannabis cures.'*—THC can kill cells in a Petri dish, but killing plated cells is entirely different from treating human cancer. |
| Shifting the burden of proof | Insisting the onus is on scientists to 'disprove' claims rather than offering any evidence or rationale for assertions made. | *'GMOs are toxic, and scientists should prove us wrong.'*—This assertion is untrue, and onus lies on those making the claim to proffer evidence for it. |

**Discreditation attempts**

| | | |
|---|---|---|
| Questioning qualification | Casting doubt on one's ability to question claims at hand. Typically, ostensible scepticism is not extended to claims supportive of the narrative. | *'This patient advocate isn't an expert, so they can't say this vaccine is safe!'*—One does not need to be an expert immunologist in this case to accurately reflect medical consensus. |
| Alleging vested interests | Claims that the speaker is compromised due to some apparent conflict of interest or that experts are otherwise lacking impartiality. | *'Who's paying you to say this?'*—Unsubstantiated allegation to deflect from absence of evidence for a narrative or claim. |
| Asserting conspiracy theory | Allegations that the scientist, physician or patient advocate is part of some conspiracy to suppress the truth or spread false information, either as a pawn or an active player. | *'She's part of a pharma cover-up to suppress natural cancer cures!'*—Appeals to conspiracy theory function to distract from lack of evidence. |

**Dubious amplification of pseudoscientific narratives**

| | | |
|---|---|---|
| Media targeting | Targeting traditional or online media outlets to amplify dubious narratives, typically bypassing gatekeepers (science/health journalists, and so on) who would otherwise be more likely to spot pseudoscience. | Pitching dubious health claims to journalists as human interest stories—This approach was successfully used by antivaccine activists to push the discredited link between autism and the MMR vaccine between 1998 and 2000. |
| Astroturfing/sockpuppeting | Use of fake social media accounts/fictitious pressure groups to provide an illusion of a wider grassroots support for a particular narrative. | Example: Accounts which spring up once an initial antifact site is blocked but which include misinformation consistent with the originator's initial social media accounts. |
| Celebrity endorsement | Celebrities and influencers can have disproportionately large impact on the perception of public even in areas where they have no relevant expertise or knowledge. | There are numerous examples of this, especially in relation to antivaccine activism, including actors and models being cited for their purported knowledge of complex health issues. |

**Malicious complaints/abuse of regulatory frameworks**

| | | |
|---|---|---|
| Poisoning the well/smear campaigns | The spreading of malicious claims regarding an individual's professional or personal conduct to undermine them or discourage others from engaging with them. | *'I've heard that doctor abuses patients.'*—Inflammatory slurs such as these are designed to discredit, and are not in any way substantiated, but calculated to invoke disgust or contempt. |
| Vexatious complaints to employers | Making calculated complaints to one's employer or threatening to do so in order to intimidate them into silence. | Exaggerated/misleading accounts of interactions with public advocates and demands to censure them, typically aimed at an individual's university or employer. |
| Vexatious complaints to regulatory bodies | Abusing procedures of professional bodies to target a researcher/medic who presents a challenge to a narrative. These may also include unwarranted freedom of information requests or vexatious parliamentary questions. | Registering complaints with a medical regulatory body against a doctor for their advocacy of evidence-based positions. Physicians especially vulnerable, as typically all complaints must be investigated, regardless of merit. |

**Intimidation**

| | | |
|---|---|---|
| Harassment/abuse | Harassment can take many forms, and personal abuse is perhaps most common. Threats (implied and direct) are often employed. | Abusive language made publicly or in direct messages, and posting of private contact details, phone numbers, addresses, and so on. |

Continued

**Table 2** Continued

| Legal threats | Legal notices and mechanisms can also be used to silence researchers questioning a narrative, from cease and desist notices to defamation claims. | Threatening to bring an advocate to court for alleged defamation, also used judiciously to limit independent investigation on pseudoscientific narratives. |
|---|---|---|
| Physical intimidation | Implicit or explicit threats of physical violence are an unfortunately potent method of intimidating scientists into silence. This includes threats of physical violence or rape (the latter usually directed at female discussants). | Stalking of private abodes, explicit threats, or actions like spitting, and so on. There are instances of security being required for scientific meetings on publicly contentious subjects, due to implications of or threats of violence. |

GMO, genetically modified organism; MMR, measles, mumps and rubella; THC, tetrahydrocannabinol.

5 and 16. Twitter is disproportionately represented, as prominence on that platform was part of the selection criteria. Other fora cited included books, documentaries, newspapers, podcasts—and in one instance criticism under parliamentary privilege. The vast majority of those surveyed (n=94, 93.1%) reported being the recipient of personal abuse of professional smears in the course of their outreach efforts. A majority (n=70, 69.3%, survey question 13) had experienced targeted abuse from at least one particular grouping. The most common groupings for negative reactions were antivaccine and complementary medicine groups. respectively (n=43 each, 42.6%) followed by dietary advocates (n=26, 25.7%), 'wellness' groups (n=17, 16.8%), religious and chronic illness groups (both n=15, 14.9%), antifluoride and autism-focused groups (both n=12, 11.9%). Others cited by three or less respondents included electromagnetic hypersensitivity factions, conspiracy theorists and anti-genetically modified organism organisations.

Figure 3 depicts types of experiences reported by participants, ranging from the relatively minor to the severe, informed by survey responses to questions 11, 12 and 14. Of participants surveyed, a majority (n=63, 62.4%) reported fallout from public engagement had caused them at least some negative mental health sequelae, including depression, anxiety and stress. Most of this was reported as minor, but considerable or significant mental health ramifications were reported by 15 respondents (14.9%). Of the participants, 20 (19.8%) were obligated to seek police advice/legal counsel as a direct result of targeted actions associated with their outreach work. Of those receiving vexatious complaints (n=39, 38.6% of all respondents), most (67.6%) felt supported or well supported by their institution, employer or professional body, while 16.2% deemed support to be neutral, with an equal number (16.2%) feeling poorly or entirely unsupported. Predictably perhaps, gender-specific abuse was far more likely to be directed at women (40% of female respondents) than men (6.8% of male respondents), with this difference being highly significant (p<0.001, calculated by Welch's t-test).

Table 3 depicts frequency of different experiences (positive and negative) reported by respondents, taken from data in survey question 8. In response to survey question 9, 29.7% (n=30) responded that they found

outreach largely rewarding, 38.6% rewarding (n=39), 29.7% mixed (n=30) and ~2% not very rewarding (n=2). Changes respondents felt would most improve outreach work are depicted in figure 4 (from survey question 20). Free-form responses to this question included: improving the media's scientific literacy (false balance and the platforming of antiscience views were repeatedly mentioned); the establishment of legal defence funds; better coordination of professional bodies; robust infrastructure on social media to report disinformation, and better support from police organisations.

## DISCUSSION

In a globalised information age, medical science can appear disconnected and aloof from those it serves to help. Educational and professional bodies (including universities and medical centres) have a unique societal role to inform their peers and public on evidence-based medicine, and a responsibility to adjust to modern communications realities. We can collectively no longer remain on the fence in supporting health information advocacy. While being mindful not to overinfer from our survey, we can use it as a basis to make some suggestions. It is vital to have proactive strategies in place to support those engaging in medical outreach, and to maintain a high calibre for public discussion. It is also crucial that those engaging in outreach are cognisant of the potential pitfalls, and afforded ample support. Given the gendered nature of much of the abuse reported, it seems likely that the hostile environment encountered online could dissuade many talented female communicators from engaging fully, to focus on but one example. It is also important to note that despite the sometimes fraught nature of medical science outreach, a majority of respondents (68.2%, n=69) found the undertakings rewarding or very rewarding. This is encouraging, but it is crucial we are aware too of the adverse effects that can arise from communicating medical science, many of which are outlined in this manuscript.

One potential weakness of the survey is the potential for ambiguous definition. Complementary medicine, for example, is typically defined as ostensible medical interventions for which there is insufficient or disconfirmatory evidence; for example, the National Science Board defines

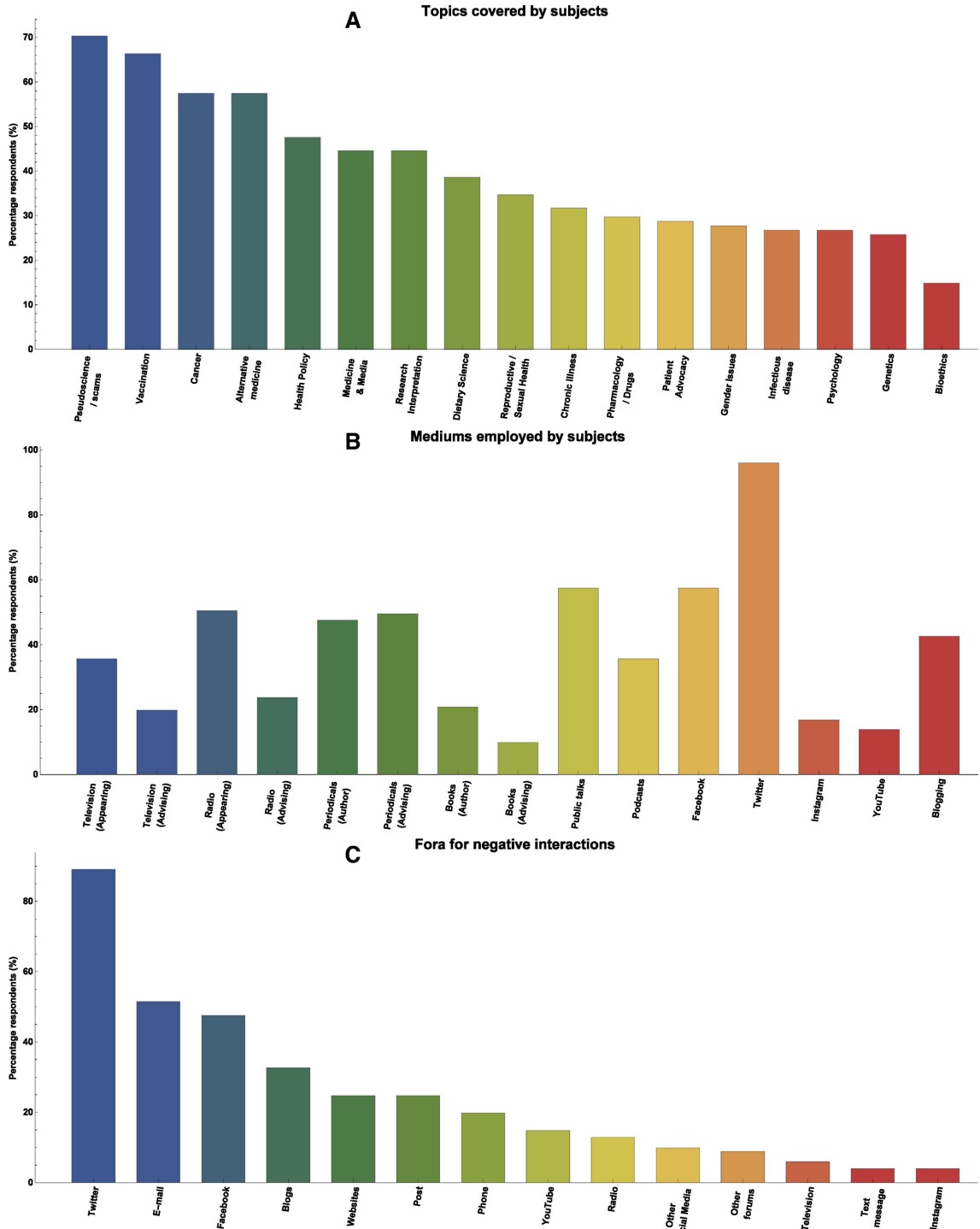

**Figure 2** (A) Topics covered by participants. (B) Channels of engagement for subjects surveyed. (C) Fora for negative interactions.

it as referring to 'all treatments that have not been proven effective using scientific methods'. As no specific definition was given in the survey text, it is possible respondents substituted their own meaning to some extent. As those surveyed were drawn from science communicators with significant media profiles however, it might be expected that their definitions were more unified than a typical respondent might be. There is also some unavoidable ambiguity with terms

such as 'abuse' and 'smears'. There is a level of subjectivity to these terms, which respondents were left to define themselves. This renders the responses potentially subjective, although the free-form responses do indicate behaviours that could be seen as objectively abusive.

There is also a serious point that must be at least considered—that advocates for medical science may on occasion engage in ill-advised tactics or unhelpful rhetoric.

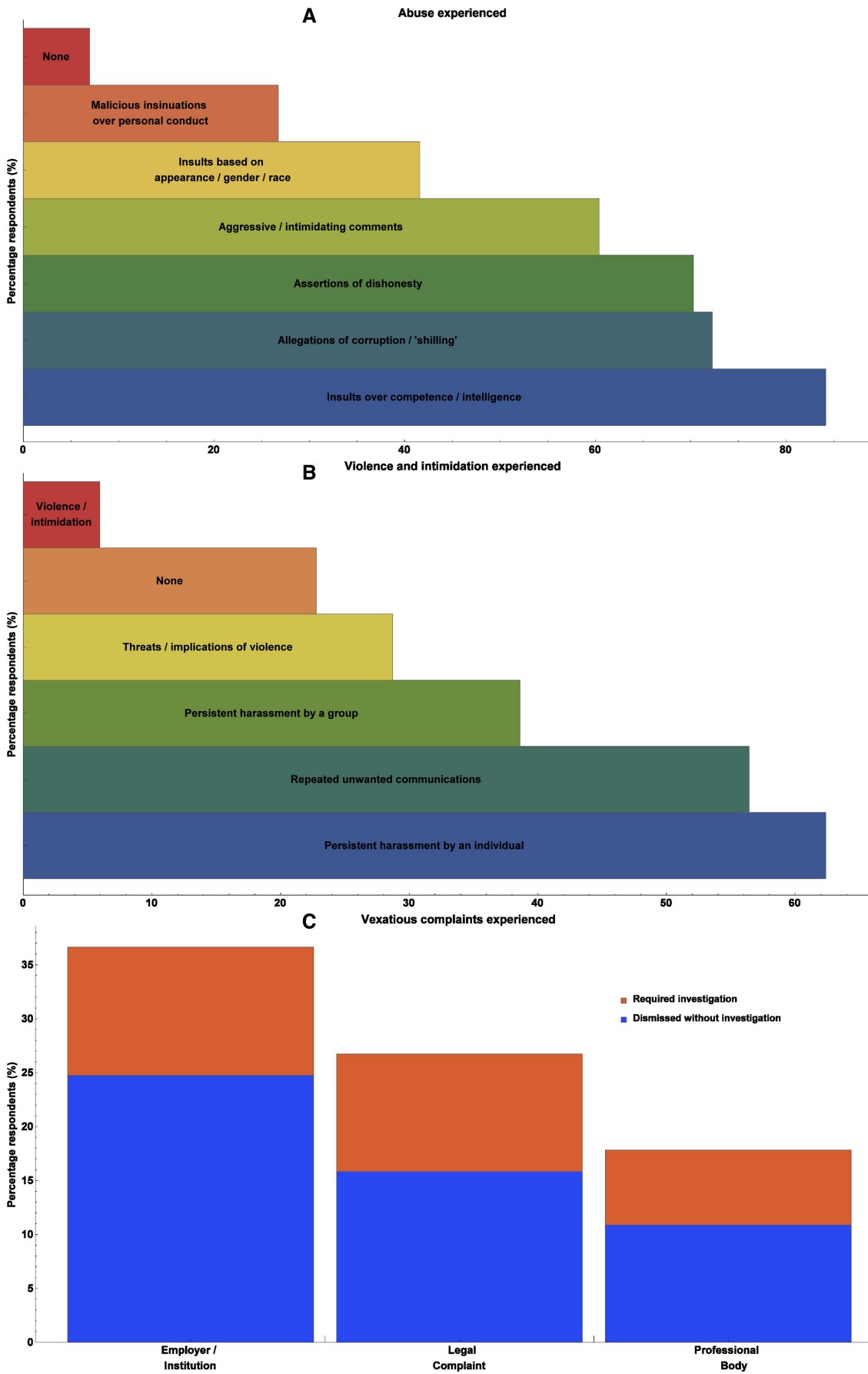

**Figure 3**   Proportion of negative experiences recorded including (A) abuse experienced, (B) violence and intimidation, and (C) vexatious complaints.

**Table 3** Frequency of experiences with outreach

| Statement | Always | Frequently | Sometimes | Infrequently | Never | Unsure |
|---|---|---|---|---|---|---|
| Engagement is mutually informative. | 6 | 31 | 52 | 10 | 2 | 0 |
| Engagement changes minds and informs. | 1 | 22 | 64 | 7 | 1 | 6 |
| My contributions are welcome and appreciated. | 1 | 56 | 39 | 3 | 0 | 0 |
| My efforts contribute to public understanding. | 4 | 47 | 49 | 1 | 0 | 0 |
| My efforts are taken in good faith. | 3 | 51 | 38 | 6 | 0 | 3 |
| My efforts feel futile. | 1 | 9 | 50 | 34 | 7 | 0 |
| Engagement takes a toll on mental health. | 2 | 12 | 44 | 26 | 16 | 0 |

Nor does one's expertise render them infallible, and it is certainly possible that advocates for science might sometimes engage in a counterproductive fashion. To ascertain this requires some context and nuance, especially for academic and medical institutions whose staff might be the subject of complaints. But rather than be reactionary, it is imperative that bodies and institutions have robust and considered policies for dealing with issues that might arise. The benefits of this are twofold; first, so that errant behaviour by members can be corrected. But equally importantly, cognisance of the reality of vexatious complaints also means that bodies and institutions can implement measures to ensure that their procedures cannot be weaponised by malicious actors. Based on the feedback to this survey and wider discussion on the topic, we offer the following suggestions to employers and professional bodies whose members might engage in the communication of medical science. While by no means comprehensive, the following guidelines might be beneficial towards establishing policy for dealing with issues that can arise.

### Suggested guidelines for professional bodies and employers

1. Educational/professional organisations must recognise a commitment to support evidence-based actions by their members. This may require oversight of such activities and investment in the governance/training resources to protect members willing to act as advocates.

2. Institutions and professional bodies should have robust measures in place to oversee communication activities associated with their members, and to make assessments as to whether individuals are communicating established facts in good faith or are contributing to undermining of facts with potential legal/reputational damage.

3. When institutions receive complaints regarding members, the subject must be afforded presumption of innocence rather than being served with reactionary and inflexible procedures, lest the institution might become an unwilling tool against science.

4. In case of disputes and complaints, competent and impartial individuals should be engaged to independent-

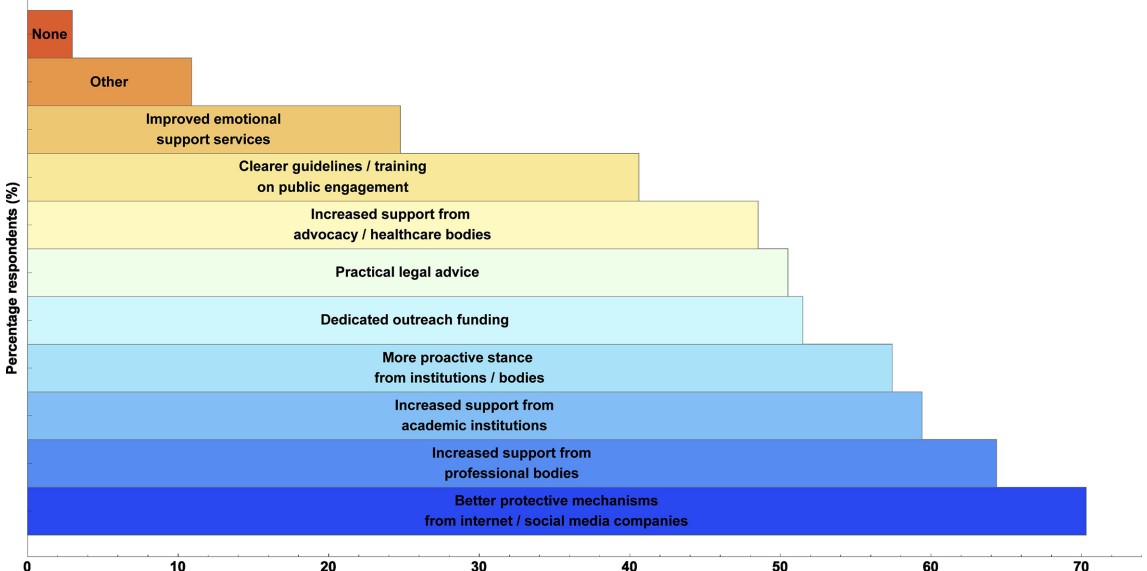

**Figure 4** Changes respondents deemed most likely to benefit medical science communication the most.

5. Coordination between press offices and those engaged in outreach would improve communication, pre-emptively identifying those likely to be targets for malicious tactics.

6. Support for those identified as being on engagement front lines should be maintained, with clear legal advice/institutional support for targeted members.

7. Organisations must be vocal in supporting public-facing members, willing to issue strong rebuttals of vexatious complaints against individuals.

8. Professional bodies and employers should strive to promote both scientific freedom of speech and to champion evidence-based advocacy.

9. When possible, those expected to engage in outreach should be trained in methods that reduce opportunities for personal and professional attacks.

## CONCLUSIONS

The question of how we best communicate health science in the modern era is an area where more research is urgently required, especially on the role of social media, and optimum ways physicians, researchers and other public-facing figures can promote good medical science and mitigate falsehoods. The suggestions herein ought to be taken as a starting point, with discussion evolving as improved evidence materialises. There are wider problems implicit in all this that those communicating science cannot tackle in isolation; social media regulation particularly is a serious issue, both in regard to the spreading of misinformation/disinformation, and with respect to procedures preventing the potential weaponisation of social media platforms. Social media platforms must ultimately be made answerable to regulatory oversight, just as every other important aspect of life is; claims of innocence are unconvincing when their business model is so clearly dependent on advertising engagement at the cost of lives. The problem of poor reporting and false balance[24] in conventional media outlets also must be considered, and there is significant scope for scientists and doctors to contribute to policy in these areas. There is ample evidence that physicians and scientists have an important role to play in combatting health disinformation, as has recently been argued by one of the authors in relation to vaccination for *British Medical Journal* opinion.[25] But equally, it is crucial that those engaging in this vital work have the requisite support from their institutions, so that deleterious consequences of laudable outreach work might be circumvented. It is increasingly clear that disinformation about medicine and illness has become ubiquitous, with severe consequences for both our collective health and public understanding of medical science. Scientists and physicians must be at the vanguard of the pushback against these dangerous falsehoods—our societal well-being depends on it.

**Acknowledgements** The authors thank the scientists, physicians and patient advocates who gave their time to discuss the issues they face in communicating science, and for sharing their insights, and the reviewers for their useful recommendations. In particular, the authors profoundly thank LJB for her invaluable input; LJB was present and helped conceive the concept from which this paper sprung. She was a passionate advocate of evidence-based science communication and especially for the HPV vaccine, championing it despite enduring a life-limiting and difficult cervical cancer diagnosis. Following a confidence crisis in the vaccine in Ireland driven by antivaccine activism, her tireless work was a substantial factor in reversing the damage wrought by disinformation. LJB passed away on 20 March 2019, but while we have lost a friend and colleague, hers is a legacy that will resonate for generations. LJB embodied precisely how vital health communication is to promoting public health; Ireland, and the world, owe her a huge debt, and we dedicate this work to her memory.

**Contributors** DRG, LJB and RO conceived the concept for this paper, and worked on survey design. DRG and RO performed the analysis and wrote the manuscript.

**Funding** DRG is supported by the Wellcome Trust (Grant number 214461/Z/18/Z).

**Competing interests** None declared.

**Patient consent for publication** Not required.

**Provenance and peer review** Not commissioned; externally peer reviewed.

**Data availability statement** Data are available upon reasonable request. All data relevant to the study are included in the article or uploaded as supplementary information. Deidentified participant data can be supplied upon reasonable request. Survey design included in standard supplementary material.

**ORCID iD**
David Robert Grimes http://orcid.org/0000-0003-3140-3278

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
