## [Reviewer comments · BMJ Open]

ARTICLE DETAILS

TITLE (PROVISIONAL)	Establishing a taxonomy of potential hazards associated with communicating medical science in the age of disinformation
AUTHORS	Grimes, David; Brennan, Laura; O'Connor, Robert

VERSION 1 – REVIEW

REVIEWER	Katie Attwell University of Western Australia, Australia Speakers fees from Merck. Travel, registration and conference fees from GSK.
REVIEW RETURNED	11-Dec-2019

GENERAL COMMENTS	Review BMJ Open The article does not appear to have been carefully proofread. There are different fonts in the document, phrases are repeated by mistake, and there are typos. This article should be subjected to machine spell checking and also read aloud by the authors before resubmission to remove these mistakes. I really like the taxonomy at Figure 1. It's fantastic. Likewise the accompanying table. However, the scholars should tell us more about the interviews they conducted to develop it. Are they published or under review with a journal? Were they a qualitative study? How were they analysed? How were people chosen to be in those interviews? How many people were interviewed? Was the study subject to ethical review? Was the taxonomy informed by any other literature? Even if the taxonomy's construction was less scientific than the authors would like to share (ie they spoke to some of their contacts and peers), their experiences in its construction should still be shared with the audience so we can better appreciate how you arrived at the design of the study reported in this article. And there are ways of writing about these methods that can demonstrate the level of rigour that still operates amongst professionals reflecting on and sharing their lived experiences. These can be investigated and employed. The methods section needs to tell us more detail about the study design and recruitment. I would also like to see the survey questions alluded to in this section and featured as a specific named table, so that we can look ahead to the table at this point. Here are some more important questions that should be answered in the methods section. How did the researchers decide who to
---

	recruit? How did they make decisions about going beyond Ireland, which may be a context they know better than others? Did they rely on people they knew from their own Twitterfeeds or go beyond this and if so, how? Could we see a map or table of the countries that participants were from? (This would go in results.) How did they recruit for the study – can we have more on this please? Did they pursue some level of gender parity? How was the survey designed and did the reviewers draw on some prior knowledge or views of their own that informed the questions? How did the researchers organise and analyse answers that were given in the open ended sections? Even good old Excel is fine, but you need to tell us what you did and how you made sense of it. The discussion section moves swiftly to a set of recommendations that institutions should adopt, but does not adequately link these back to the findings. A linking section should be added in which the authors pull out their key findings and consider their significance. This could include the prevalence of particular problems (even if the N is small in their exploratory study) or the meanings and problems associated with others. For example, high levels of gendered abuse could mean that more women are deterred from persisting with science communication than men. Or the numbers of people suffering mental health issues could lead to attrition of experienced science communicators, leaving the field more open to the charlatans. It is the authors' job to think through more of these implications and communicate them effectively, so that we can then see how the recommendations address them. This linking section could be successfully done via a numbering or labelling process that then allows the authors to refer back the recommendations to addressing particular deficits of current practice. (ie recommendation 4 addresses problem H, or whatever.) Without this linking, there is a risk that the list of recommendations simply hangs there as smart ideas that could well have been cooked up independently of this survey. Ultimately, I'd like to see this article published, and I look forward to seeing the author's improvements following review.
--	---

REVIEWER	Dónal O'Mathúna The Ohio State University, USA, and Dublin City University, Ireland
REVIEW RETURNED	27-Dec-2019

GENERAL COMMENTS	This article is an important step in examining the negative impact of being involved in advocating for medical information and evidence-based decisions. However, I have concerns about the reported results, especially around the completeness of what is reported based on comparing the figures, text and survey questions. I think some of these items can be addressed and remedied by the authors, but it will require further elaboration on the methods and results. However, it may be that there are limitations in the original survey that will be difficult to overcome. I'll address the issues as they arise in the manuscript.
--

Introduction: at the end of the second paragraph (page 2, line 30), “overwhelming scientific consensus” is mentioned. However, this can be a controversial term. How exactly is it defined here? This is especially important in relation to discussions of alternative medicine, given how broad and varied this topic is in general. Within alternative medicine, there are a large variety of therapies and remedies, some with more scientific evidence in support than others. This means that clear definitions of what are being including in each category are necessary.

The issue of definitions is particularly important with the survey reported because the questions used many terms that were not defined. For example, in Q.9, vexatious complaints are defined, but no other terms are similarly defined. Ones like “smears” and “abuse” are very likely to be interpreted differently by different people, and this weakens the usefulness of the survey.

On the bottom of p. 3, “weaponising” is used where this would seem to be a very extreme term for what may be happening. Besides, the findings from your own survey do not support the claims that professional bodies are frequently being used against the science advocates. Most of them find their bodies to be supportive. I would find a different term, and likewise not use “weaponising” later on p. 9 because it is such an inflammatory (and ambiguous) term.

Methods: The taxonomy is very interesting and useful. However, we need more details on how this was developed as practically none are given. Who was interviewed? How were the data analysed?

For the survey, further details are needed about where the respondents were drawn from. This is important given that the sample was not randomised and therefore is prone to being biased depending on who was asked. The authors acknowledge that the results are not generalizable, but further transparency on where the sample came from would be good.

It would also be good to know how the survey was developed. Was this based on the taxonomy or feedback from the interviewees? Also, there was no mention as to whether or not the survey was validated or if it was piloted and revised. This information is important to assess how well the survey captures reliable findings.

The authors mention that the survey was exempt from HRA/HRB approval. However, this is not the same as research ethics approval. It would be good to mention if this was sought from a research ethics committee or IRB, and if they gave exemption. Given that qualitative interviews appear to have been used for the taxonomy, and that the survey could also have been deemed to require ethics submission, further information is needed on these points.

Results: I had a lot of confusion here. Part of this might be because the pages with the bar charts did not have labels. I’m assuming page 12 was Figure 2 and page 13 was Figure 3. The text on p. 7 refers to Figure 2, and then gives numbers relating to the topic groupings that caused negative experiences. The numbers seem to come from Q. 13, but Figure 2(a) is from Q. 4. This needs to be made clear. However, the terminology also needs to be kept consistent. For example, Q. 13 asked about “targeted abuse” while the text on p. 7 reports “ire”. These are not the same thing. Also, Q. 4 asked about “pseudoscience/scams” while Figure 2(a) is labelled “pseudoscience/health scams.” Other labels are not copied accurately and should be revised. The same applies to all figures and sub-figures.

	Another question I have is how the results reported were selected. Admittedly, the focus is on the negative experiences, but the survey asked other questions and these answers are not reported. I'm concerned that the selective reporting gives an unbalanced perspective. For example, in reporting the results of Q.13 above, the number of "I have not encountered this" responses is not provided even though every other category is. With Figure 3(b), assuming it comes from Q.12, the number of responses to the rumours question is not provided. The answers to Q.8 and 10 contain positive experiences as well as negative, and these results would be valuable to compare with the negative. If a high percent of respondents believe their efforts are effective and valuable, in spite of the negative aspects, that is relevant to the overall picture. If the positive experiences are low, that is also relevant and points to a major problem. Similarly, the answers to Q.20 are directly relevant to the proposed guidelines, yet the results are not included. If these answers support the proposed guidelines, that would support the authors' arguments. At the very least, we need to be told about how the answers were selected to be reported, and especially if this was an a priori aspect of the methodology. Discussion: the data on the types of negative experiences people have had is interesting. However, as presented, it would be helpful if there could be some discussion about the frequency of such negative experiences. From the survey questions, I'm not sure if anything can be reported about this, but maybe it can from the interviews. Either way, this is a serious limitation to the survey which needs to be acknowledged. As reported, there seems to be no way to determine if these sorts of events happen once every few years, or more regularly. This makes the study more like the way people leave reviews online – we can only assume that people are providing accounts of their worse and best experiences. If these problems occur infrequently, then the list of suggested guidelines might not need to be prioritised. Or maybe they do. My point is that the reader can't tell from what is reported, and this should at least be acknowledged. The limitations of the survey methods require some discussion of alternative factors in the reported experiences. For example, the assumption seems to be made in the article that the negative experiences are the results of the misinformation side of the debate. Is it not possible that the scientific advocates may have contributed to unhealthy debates? Might the science advocates also have presented things in ill-advised ways so that the debate deteriorated into a negative experience? I have witnessed such types of interactions where both sides were at fault, yet no mention is made of this possibility and the implicit suggestion is that the problems lie solely with the misinformation side. In fact, in some bioethics debates that I have participated in, it has been the science advocates who have committed some of the fallacies listed in the taxonomy. The article would do better to acknowledge that science communicators can and do make mistakes. This might warrant a guideline on providing training and feedback to science communicators. This might be implicit in some of them, but the training needs to go beyond knowledge of content to knowledge of the principles of good argumentation and the avoidance of fallacies by science advocates. One small point on the references is that on p. 9, the authors state that "the authors" have argued in the British Medical Journal. However, the reference shows that only one of the authors wrote the piece which is in BMJ's blogging site, BMJ Opinion. This is a
--	---

	different sort of publication to a peer-reviewed article in BMJ, and that should be made explicit for the readers. All of the points above should be discussed in a Limitations section at the end of the article. While some limitations are mentioned, this needs to be developed further. This survey is interesting, but has more serious limitations than are presently discussed. In the interests of the articles topic on scientific communication, these points need to be addressed more fully. There are many, many typos which need to be corrected. I have listed some of the more obvious ones below, but the article needs a careful and thorough proofreading. Page 1, line 34: there should be an “or” between employers and professional, as legal intimidation is not covered by complaints. Page 1, line 38: / should not have spaces on either side Page 2, line 12: the dash within “endeav-ours” should be removed. This sort of typo occurred a few times. Page 2, line 18: “media gures”. Is “guru” what is meant here? The term gure is used several places, and I’ve never seen it before. Page 2, line 34: replace “their” with “the” Page 2, line 39: delete “cases” after measles as it was already stated before the word. Page 2, line 39: replace “death” with “deaths” Page 2, line 42: “to anti-vaccine conspiracy” is repeated Page 2, line 43: I believe “not” is missing from before “to vaccinate” Page 2, line 53-54: these 2 sentences are highly repetitive Page 2, line 57: replace “half” with “halve” Page 3, line 19: fix “ctions” Page 3, line 40: replace “attacking” with “attack” Page 4, line 42: various letters are missing, making this unintelligible. Page 4, line 42: The names of specific tables and figures should be capitalised: Table 1, etc. There are many other typos on the following pages, but a thorough proofreading should correct these.
--	--

VERSION 1 – AUTHOR RESPONSE

Replies to Reviewer 1

1. *“The article does not appear to have been carefully proofread. There are different fonts in the document, phrases are repeated by mistake, and there are typos. This article should be subjected to machine spell checking and also read aloud by the authors before resubmission to remove these mistakes”*

Our sincere apologies for this oversight – the document was written in latex, and upon conversion parts became corrupted. We have rectified these slip-ups now.

2. *“I really like the taxonomy at Figure 1. It’s fantastic. Likewise the accompanying table. However, the scholars should tell us more about the interviews they conducted to develop it. Are they published or under review with a journal? Were they a qualitative study? How were they analysed? How were people chosen to be in those interviews? How many people were interviewed? Was the study subject to ethical review? Was the taxonomy informed by any other*

literature? Even if the taxonomy's construction was less scientific than the authors would like to share (ie they spoke to some of their contacts and peers), their experiences in its construction should still be shared with the audience so we can better appreciate how you arrived at the design of the study reported in this article. And there are ways of writing about these methods that can demonstrate the level of rigour that still operates amongst professionals reflecting on and sharing their lived experiences. These can be investigated and employed."

This is an excellent point – to answer specifically, the interviews stemmed not from an academic examination, but rather grew from experiences shared in private member groups by medical scientific communicators. This formed the basis of the set questions for the survey. We also added sections for participants to share their own experiences, if they were willing. This was the genesis of the idea, especially when members of these international groupings were targeted by specific groups. Finding a shortage of literature on the topic, we decided to address this by the means at our disposal. We agree we should clarify this process. On that note, we have added the following text to the manuscript, and rewritten the methods section entirely to encompass this, and to clarify our selection process (see reviewer 2 also).

"The main recruitment fora for this study was online discussion groups for physicians, scientists, and patient advocates communicating aspects of medical science to the general public across social and traditional media. At the height of the Irish HPV vaccine confidence crisis, several physicians and scientists (based in Ireland and the UK) in these groups sought advice for negative experiences, including threats and malicious complaints to their employers and professional bodies, all of which were eventually dismissed. Group members across Europe echoed similar experiences in communicating vaccination science and in other health issues, and almost identical adverse reports came from colleagues across the America and Africa. Informal interviews were conducted on foot of this in these fora to identify common issues, as to the author's knowledge there is no existent literature on the topic.

On basis of these interviews and related fora discussions, a survey was created, including free-form sections where subjects were free to expand on their own experiences. The wording of this survey is included in the supplementary material. The selection criteria was specifically for those communicating medical science both on social media (defined as having over 1000 followers on twitter) and in mainstream channels (defined as invited appearances on public television, radio, and / or in the form of newspaper articles & invited comment). From this, 142 individuals world-wide (from across Europe, America, Africa, and Asia) working predominantly in the English language were identified as fitting the criteria and invited to partake, of whom 101 (68.2%) responded. This survey was undertaken for indicative purposes and was collected from a non-randomised group with no expectation of transferable findings. Accordingly, it was thus exempt from requiring HRA / HRB approval, though informed consent was sought and obtained prior to subjects partaking, with all data appropriately anonymised. Subject details are given in table 1.

Patient and Public Involvement

As patient advocates play a substantial role to play in combatting misinformation on medical issues, several who met the inclusion criteria were invited to take part, with 15.8% of respondents being patient advocates. "

Please note also that the taxonomy table and figures have been moved to results to avoid potential confusion. Finally, to show how the survey fed into the taxonomy, we have added some quotes from subjects who were content to share, as detailed in response to point 3 below. As the survey findings are not generalisable in this instance given the selection criteria, it did not fit the Health Research Authority (HRA) definition of research, and Queen's University Belfast deemed it to not require ethics committee oversight, as explained in reviewer 2 reply 7.

3. *"The methods section needs to tell us more detail about the study design and recruitment. I would also like to see the survey questions alluded to in this section and featured as a specific named table, so that we can look ahead to the table at this point. Here are some more important questions that should be answered in the methods section. How did the researchers decide who to recruit?"*

As mentioned in reply to point 2 above, we have expanded somewhat on how recruitment selection was made, and the context for it. The survey is relatively long at over 20 questions, with free-form text sections and many aspects with multiple choice, so we have for the most part confined this to the supplementary material for brevity and clarity.

We have however decided in light of the reviewers comments to expand on how the taxonomy was informed, and have added sample responses from the participants into the results to illustrate the experiences that formed it. The added text to the results section is:

"In addition to survey questions on known problems, participants were also invited to expand on noteworthy negative situations they may have encountered while engaging in health outreach. This section was entirely optional, and 53 participants (52.5% of subjects) opted to share their experiences. Example responses are shown below; please note that some responses have been edited or partially redacted to exclude potentially identifying information and preserve anonymity.

"Accusations - including by one Senator - that [we are] uncaring, dismissive, neglectful, arrogant, or paid by pharma companies when advocating for vaccines." (Misrepresentation)

"I find my expertise is questioned - this often seems to be when men find it difficult to accept women with intelligence and qualifications. Sexist insults are a typical go-to response." (Discreditation)

"The worst one that hurt me professionally and personally was that activists gathered my emails using [Freedom of Information Requests] and handed chosen packets of them with a story to different reporters." (Misrepresentation / Discreditation / Dubious Amplification)

"Persistent negative comments on twitter; usually it doesn't last long but it can feel very intense while it's happening!" (Intimidation)

"I have been served with a SLAPP lawsuit in order to silence my outreach work. Frequently receive harassing emails, malicious comments made on blog." (Malicious Complaints)

"Social media co-ordinated intimidation, implied threats of legal action (for defamation). Mocking, undermining, condescension and attacks for being an industry shill, although.. I am just a patient advocate. Being called a liar, that I never had cancer, that I deserved cancer due to my attitude, that I have been mutilated by conventional medical treatment, and that I am no longer a woman (having had mastectomy for cancer). That my cancer will return and I deserve that." (Dubious Amplification / Misrepresentation / Discreditation / Malicious complaints)

"I have had anti-vaccine organizations and individuals attempt to prevent my public appearances and have been the subject of numerous online smear campaigns accusing me of being 'a shill for Big Pharma' etc." (Discreditation / Dubious Amplification)

"Those who attack me very frequently try to do it by targeting me at my job, sending bogus complaints to my bosses and the university. From my observation, that is the go-to attack, the first thing these groups do." (Malicious Complaints)

"I had to contact the police, who visited the person who was harassing me. I also involved social services. We bought a CCTV to monitor our front door after a strange envelope was hand delivered. The person involved has targeted several people before and continues to target individuals who advocate vaccination." (Intimidation)

"Abuse and accusations of corruption are the most common adverse reaction I get. Sometimes a particular group petition one's employer and try to create trouble for them. I have been lucky in the past when this happened to have had supportive universities who appreciate my outreach work. I have in the past had slightly unhinged individuals writing rambling, implicitly threatening letters to my office which ultimately required police intervention." (Discreditation / Malicious Complaints / Intimidation)

"The worst are gendered insults (being called a cunt, etc.) and rape/death threats. I have had one empty legal threat that was widely publicized." (Intimidation / Malicious Complaints)

“Regular threats to sue for defamation.” (Malicious Complaints)

“Attempts to get me fired, public records act requests for emails, verbal attacks on my children.” (Malicious Complaints, Intimidation)

“One of the most unpleasant things is that certain people or groupings will use very underhanded tactics to respond to perceived criticism. If they can't refute the science, it isn't uncommon for them to go after you personally, alleging all manner of things to anyone who'll listen; that you're incompetent, or unethical, or perverted. It seems they throw things wildly to see what sticks, but it can be extraordinarily unpleasant to endure.” (Dubious Amplification, Discreditation)

“My main concern has been obsessed individuals who declare their enmity and seem to be unconstrained by civil norms.” (Intimidation)

“Homeopathy advocates looked up my LinkedIn profile and called my employer to complain about my comments on the radio. My employer did not support me and I ended up having to stop the activity I had been planning.” (Misrepresentation / Malicious Complaints)

“Being threatened with physical violence” (Intimidation)

“A delusional supporter of [an individual] I wrote about accused me and my lawyer of stalking him and killing his in-laws. He sent accusing emails to the faculty of my school and all the police departments in my state. [They] also accused me of being a terrorist and complained about me to the FBI's Terrorism Joint Task Force. That gave me many nervous, sleepless nights.” (Discreditation / Malicious Complaints)

“Death threats received, employer unhelpful, sorted myself” (Intimidation)

“I haven't experienced many negative encounters because I would say I am only lightly involved in public engagement. However the reason I don't become more heavily involved in this area is fear of this kind of abuse and vexatious complaints to my employer or regulatory body.” (Malicious Complaints)

Based upon this and survey responses, a non-exhaustive taxonomy was constructed detailing common experiences of those communicating medical science to the public. While non-exhaustive, it forms a useful basis for more systematic investigation. Adversarial experiences in communicating medicine were broadly stratified into five distinct classes, illustrated graphically in figure 1, with these sub-types detailed in detail in table 2. Participant details are given in table 2. Topics covered / channels of engagement are depicted in figure 2(a). “

- 4. “How did they make decisions about going beyond Ireland, which may be a context they know better than others? Did they rely on people they knew from their own Twitterfeeds or go beyond this and if so, how? Could we see a map or table of the countries that participants were from? (This would go in results.) How did they recruit for the study – can we have more on this please? Did they pursue some level of gender parity? How was the survey designed and did the reviewers draw on some prior knowledge or views of their own that informed the questions? How did the researchers organise and analyse answers that were given in the open ended sections? Even good old Excel is fine, but you need to tell us what you did and how you made sense of it.”*

Analysis of data was relatively straight-forward – an author simply wrote a script to pull in the raw numbers from different question responses into MATLAB and to query them. For the most part, this is what we report. We have now added open-ended answers to the manuscript, as discussed in the previous section. We cannot alas give the locations of survey respondents, as we opted not to ask this question lest it identified a well-known person in the field by mistake, as some jurisdictions have a smaller community of medical science communicators. More than that, many are pan-national in operation. We have however mentioned in the text above which regions communicators invited chiefly operated, as detailed above.

5. *“The discussion section moves swiftly to a set of recommendations that institutions should adopt, but does not adequately link these back to the findings. A linking section should be added in which the authors pull out their key findings and consider their significance. This could include the prevalence of particular problems (even if the N is small in their exploratory study) or the meanings and problems associated with others. For example, high levels of gendered abuse could mean that more women are deterred from persisting with science communication than men. Or the numbers of people suffering mental health issues could lead to attrition of experienced science communicators, leaving the field more open to the charlatans. It is the authors’ job to think through more of these implications and communicate them effectively, so that we can then see how the recommendations address them. This linking section could be successfully done via a numbering or labelling process that then allows the authors to refer back the recommendations to addressing particular deficits of current practice. (ie recommendation 4 addresses problem H, or whatever.) Without this linking, there is a risk that the list of recommendations simply hangs there as smart ideas that could well have been cooked up independently of this survey.”*

This is a lovely idea, but one which we feel if implemented precisely as above would have to be confined to supplementary material, as we are already significantly beyond the recommended word limit. However, we feel that with the addition of the personal anecdotes (described in section 3) which informed the taxonomy, the rationale for these recommendations becomes clearer. We very much like the reviewers suggestions regarding potential consequences, and we have expanded the discussion in light of this – please see replies to reviewer 2 for expanded discussion, including new text on the likelihood of the gendered abuse prevalent in discussions online driving away skilled female science and medical communicators.

Replies to Reviewer 2

1. *“Introduction: at the end of the second paragraph (page 2, line 30), “overwhelming scientific consensus” is mentioned. However, this can be a controversial term. How exactly is it defined here? This is especially important in relation to discussions of alternative medicine, given how broad and varied this topic is in general. Within alternative medicine, there are a large variety of therapies and remedies, some with more scientific evidence in support than others. This means that clear definitions of what are being including in each category are necessary. “*

This is a very pertinent point, and one we address in reply to this reviewer, point 2.

2. *“The issue of definitions is particularly important with the survey reported because the questions used many terms that were not defined. For example, in Q.9, vexatious complaints are defined, but no other terms are similarly defined. Ones like “smears” and “abuse” are very likely to be interpreted differently by different people, and this weakens the usefulness of the survey.”*

Regarding this point and the previous, we believe the reviewer has identified a limitation of survey data that is worth expounding upon. Alternative medicine is most often defined as ostensible medical practices for which there is insufficient evidence or disconfirmatory data. It is however possible that survey respondents might substitute their own meaning to some extent, although as the sample survived were drawn from science communicators with significant media profiles (see replies to reviewer 1, points 2 & 3), it might be expected their definitions would be relatively unified for the most part. Regarding the issues of “smears” and “abuse” we have deliberately let respondents define these terms themselves, which might be a weakness. However, as can now be seen from survey respondent’s direct words which we have since added (see reviewer 1 reply 3) that these terms have more legitimacy. To acknowledge these weaknesses, we have added the following to the discussion.

“One potential weakness of the survey is the potential for ambiguous definition. Alternative medicine, for example, is typically defined as ostensible medical interventions for which there is insufficient or

disconfirmatory evidence; for example, the National Science Board define it as referring to “all treatments that have not been proven effective using scientific methods”. As no specific definition was given in the survey text, it is possible respondents substituted their own meaning to some extent. As those surveyed were drawn from science communicators with significant media profiles however, it might be expected that their definitions were more unified than a typical respondent might be. There is also some unavoidable ambiguity with terms such as “abuse” and “smears”. There is a level of subjectivity to these terms, which respondents were left to define themselves. This renders the responses potentially subjective, although the free-form responses do indicate behaviours that could be seen as objectively abusive.”

- 3. “On the bottom of p. 3, “weaponising” is used where this would seem to be a very extreme term for what may be happening. Besides, the findings from your own survey do not support the claims that professional bodies are frequently being used against the science advocates. Most of them find their bodies to be supportive. I would find a different term, and likewise not use “weaponising” later on p. 9 because it is such an inflammatory (and ambiguous) term.”*

As has now been made clearer in the text by reference to survey respondents' words, there is unfortunately a number of cases where this occurs. However, we recognise the reviewer's point here, and had added the qualifier of “potentially” so temper it somewhat. We do not however wish to downplay the experiences of those who have suffered this, of which we can provide several accounts if requested.

- 4. “Methods: The taxonomy is very interesting and useful. However, we need more details on how this was developed as practically none are given. Who was interviewed? How were the data analysed?”*

This has been clarified in this iteration. Please see reply to reviewer 1, points 2-4 inclusive.

- 5. “For the survey, further details are needed about where the respondents were drawn from. This is important given that the sample was not randomised and therefore is prone to being biased depending on who was asked. The authors acknowledge that the results are not generalizable, but further transparency on where the sample came from would be good.”*

This has been clarified in this iteration. Please see reply to reviewer 1, points 2-4 inclusive.

- 6. “It would also be good to know how the survey was developed. Was this based on the taxonomy or feedback from the interviewees? Also, there was no mention as to whether or not the survey was validated or if it was piloted and revised. This information is important to assess how well the survey captures reliable findings.”*

This has been clarified in this iteration. Please see reply to reviewer 1, points 2-4 inclusive.

- 7. “The authors mention that the survey was exempt from HRA/HRB approval. However, this is not the same as research ethics approval. It would be good to mention if this was sought from a research ethics committee or IRB, and if they gave exemption. Given that qualitative interviews appear to have been used for the taxonomy, and that the survey could also have been deemed to require ethics submission, further information is needed on these points.”*

When work on this project began, the lead author was at Queen's University Belfast. The research governance office were approached, and clarified that “surveys being conducted are not considered to be research” and that “ethical approval would not be required”. The HRA research tool also indicated specific ethical approval was not required. The term interviews was potentially misleading, as the initial impetus was actually comments in members forums for medical science communicators. This has since been clarified in this iteration (see reply to reviewer 1, point 2).

- 8. “Results: I had a lot of confusion here. Part of this might be because the pages with the bar charts did not have labels. I'm assuming page 12 was Figure 2 and page 13 was Figure 3. The text on p. 7 refers to Figure 2, and then gives numbers relating to the topic groupings that caused negative experiences. The numbers seem to come from Q. 13, but Figure 2(a) is from*

Q. 4. This needs to be made clear. However, the terminology also needs to be kept consistent. For example, Q. 13 asked about “targeted abuse” while the text on p. 7 reports “ire”. These are not the same thing. Also, Q. 4 asked about “pseudoscience/scams” while Figure 2(a) is labelled “pseudoscience/health scams.” Other labels are not copied accurately and should be revised. The same applies to all figures and sub-figures.”

We apologise for this – part of the issue here is the BMJ Open policy of not embedding figures in the main text for review. To circumvent this, we have redone figure 2, and cleaned up the terminology. This is shown overleaf for clarity. We have now clarified the precise survey questions used to inform different figures, as shown below:

“Participant details are given in table 2. Topics covered / channels of engagement and fora for abusive interactions are depicted in figure 2, informed by survey questions 4,5, and 16. Twitter is disproportionately represented, as prominence on that platform was part of the selection criteria. Other fora cited included books, documentaries, newspapers, podcasts - and in one instance criticism under parliamentary privilege. The vast majority of those surveyed (N = 94, 93.1%) reported being the recipient of personal abuse of professional smears in the course of their outreach efforts. A majority (N = 70, 69.3%, survey question 13) had experienced targeted abuse from at least one particular grouping. The most common groupings for negative reactions were anti-vaccine and alternative medicine groups respectively (N = 43 each, 42.6%) followed by dietary advocates (N = 26, 25.7%), ‘wellness’ groups (N = 17, 16.8%), religious and chronic illness groups (both N = 15, 14.9%), anti- fluoride and autism focused groups (both N =12, 11.9%). Others cited by 3 or less respondents included electromagnetic hypersensitivity factions, conspiracy theorists and anti-GMO organisations.”

Figure 2 (a) Topics covered by participants (b) channels of engagement for subjects surveyed (c) Fora for negative interactions

9. *“Another question I have is how the results reported were selected. Admittedly, the focus is on the negative experiences, but the survey asked other questions and these answers are not reported. I’m concerned that the selective reporting gives an unbalanced perspective. For example, in reporting the results of Q.13 above, the number of “I have not encountered this”*

responses is not provided even though every other category is. With Figure 3(b), assuming it comes from Q.12, the number of responses to the rumours question is not provided. The answers to Q.8 and 10 contain positive experiences as well as negative, and these results would be valuable to compare with the negative. If a high percent of respondents believe their efforts are effective and valuable, in spite of the negative aspects, that is relevant to the overall picture. If the positive experiences are low, that is also relevant and points to a major problem. Similarly, the answers to Q.20 are directly relevant to the proposed guidelines, yet the results are not included. If these answers support the proposed guidelines, that would support the authors' arguments. At the very least, we need to be told about how the answers were selected to be reported, and especially if this was an a priori aspect of the methodology."

Question 13 is only shown in the text, and does specify that approximately 30% had not experienced abuse from a particular group. Figure 3(b) does not come from question 12 – question 3(a) does, and the number is 27. To clarify this, we've added the following text to the results section:

"Figure 3 depicts types of experiences reported by participants, ranging from the relatively minor to the severe, informed by survey responses to questions 11, 12, and 14."

We take on board the reviewer's other comment about negativity – obviously we are interested in negative experiences so we may circumvent them, but it's important to frame this properly. Please see our response to point 10 for discussion on this. We also agree with the reviewer that results from Q20 ought to be made clear, and elaborated upon in discussion and result. Figure is depicted below:

New figure 4 (based on results from Q20)

10. "Discussion: the data on the types of negative experiences people have had is interesting. However, as presented, it would be helpful if there could be some discussion about the frequency of such negative experiences. From the survey questions, I'm not sure if anything can be reported about this, but maybe it can from the interviews. Either way, this is a serious limitation to the survey which needs to be acknowledged. As reported, there seems to be no way to determine if these sorts of events happen once every few years, or more regularly. This makes the study more like the way people leave reviews online – we can only assume that people are providing accounts of their worse and best experiences. If these problems occur infrequently, then the list of suggested guidelines might not need to be prioritised. Or maybe

they do. My point is that the reader can't tell from what is reported, and this should at least be acknowledged."

We agree this can be done better – firstly, in the results section, we have now added the available frequency data from question 8 pertinent to both positive experiences and negative experiences, and question 9. This encompasses a new table and line in the result text.

Table 3: Frequency of experiences with outreach

Statement	Always	Frequently	Sometimes	Infrequently	Never	Unsure
Engagement is mutually informative	6	31	52	10	2	0
Engagement changes minds & informs	1	22	64	7	1	6
My contributions are welcome & appreciated	1	56	39	3	0	0
My efforts contribute to public understanding	4	47	49	1	0	0
My efforts are taken in good faith	3	51	38	6	0	3
My efforts feel futile	1	9	50	34	7	0
Engagement takes a toll on mental health	2	12	44	26	16	0

"Table 3 depicts frequency of different experiences (positive and negative) reported by respondents, taken from data in survey question 8. In response to survey question 9, 29.7% (N=30) responded that they found outreach largely rewarding, 38.6% rewarding (N=39), 29.7% mixed (N=30), and ~2% not very rewarding (N=2). Changes respondents felt would most improve outreach work is depicted in figure 4 (from survey question 20). Free-form responses to under category of "other" included Improving media science literacy (false balance and issues with platforming of anti-science views were repeatedly mentioned), legal defence funds, better coordination of professional bodies, robust infrastructure on social media to report disinformation, and better support from police organisations."

We also added the following text to the discussion for clarity on the over-all positives.

"Given the gendered nature of much of the abuse reported, it seems likely that the hostile environment encountered online could dissuade many talented female communicators from engaging fully, to focus on but one example. It is also important to note that despite the sometimes fraught nature of medical science outreach, a majority of respondents (68.2%, N = 69) found the undertakings rewarding or very rewarding. More can be done to avoid common pitfalls of this endeavour, however, much of which is outlined in this manuscript."

- 11. "The limitations of the survey methods require some discussion of alternative factors in the reported experiences. For example, the assumption seems to be made in the article that the negative experiences are the results of the misinformation side of the debate. Is it not possible that the scientific advocates may have contributed to unhealthy debates? Might the science advocates also have presented things in ill-advised ways so that the debate deteriorated into a negative experience? I have witnessed such types of interactions where both sides were at fault, yet no mention is made of this possibility and the implicit suggestion is that the problems lie solely with the misinformation side. In fact, in some bioethics debates that I have participated in, it has been the science advocates who have committed some of the fallacies listed in the taxonomy. The article would do better to acknowledge that science communicators can and do*

make mistakes. This might warrant a guideline on providing training and feedback to science communicators. This might be implicit in some of them, but the training needs to go beyond knowledge of content to knowledge of the principles of good argumentation and the avoidance of fallacies by science advocates”

This is an excellent point, and we have alluded to it in the suggested guidelines regarding disputes 24 in the guidelines, at least implicitly acknowledging advocates can be wrong. The ones surveyed here had substantial media profiles for the most part, and perhaps that means them less likely to engage in off-putting tactics, but certainly on the wider discussion especially across social media we must address this. It would also tie the guidelines together a lot better. We have accordingly added the following text to the discussion.

“There is also a serious point that must be at least considered – that advocates for medical science may on occasion engage in ill-advised tactics or unhelpful rhetoric. Nor does one’s expertise render them infallible, and it is certainly possible that advocates for science might sometimes engage in a counterproductive fashion. To ascertain this requires some context and nuance, especially for academic and medical institutions whose staff might be the subject of complaints. But rather than be reactionary, it is imperative that bodies and institutions have robust policy for dealing with issues that might arise, both so they can correct errant behaviour by members and so that they cannot be weaponised by malicious complaints. Based on the feedback to this survey and wider discussion on the topic, we offer the following suggestions to employers and professional bodies whose members might engage in the communication of medical science. While by no means comprehensive, the following guidelines might be beneficial towards establishing policy for dealing with issues that can arise.”

12. *“One small point on the references is that on p. 9, the authors state that “the authors” have argued in the British Medical Journal. However, the reference shows that only one of the authors wrote the piece which is in BMJ’s blogging site, BMJ Opinion. This is a different sort of publication to a peer-reviewed article in BMJ, and that should be made explicit for the readers.”*

A fair point, apologies if this was misleading. It has now been remedied in the text.

13. *“All of the points above should be discussed in a Limitations section at the end of the article. While some limitations are mentioned, this needs to be developed further. This survey is interesting, but has more serious limitations than are presently discussed. In the interests of the articles topic on scientific communication, these points need to be addressed more fully.”*

We agree wholeheartedly. We have combined these points in the discussion (see replies to points 10 & 11).

14. <Typos>

Thank you for listing these – as specified in reply to reviewer 1,1 many were introduced in porting the document from latex to word. These have now been corrected.

VERSION 2 – REVIEW

REVIEWER	Katie Attwell University of Western Australia
REVIEW RETURNED	18-Feb-2020

GENERAL COMMENTS	I am satisfied with the revisions to this article and believe the authors to have addressed key issues raised by me and the other reviewer. I have no further changes to request.
---

REVIEWER	Dónal O’Mathúna
-----------------	-----------------

	The Ohio State University
REVIEW RETURNED	26-Feb-2020

GENERAL COMMENTS	Overall, this manuscript has improved in many ways. Thank you to the authors for taking the reviewers' comments on board and doing quite a bit of work. I think the result is a greatly improved article that will give readers a clearer picture of the work they have done in this area. Well done. There are still some places where I think sentences could be clarified. Those are noted below. In addition, there continue to be many typos – some of which were in the previous version of the article. It may be that the authors are too familiar with their work to spot these, so I would recommend having someone else with good writing skills proofread the article again. I have highlighted typos which I noticed in the pdf of the manuscript, although in some places I couldn't do this because of the watermark. Also, throughout the document, the forward slash (/) has spaces on each side. This is not standard, and the spaces should be removed, to leave, for example, "and/or". p. 4, line 48: the sentence ends with saying something will be counteracted. In context, this refers to the experiences of communicating medical science. But why should these experiences be counteracted? It's not clear that they are all negative, and the results find positive and negative. This sentence should be rewritten. p. 5, line 23: I think the authors mean "The participant selection criteria", and if so, this word should be added. Note also that criteria is plural and therefore should be followed by "were". p. 5, line 26: What does "From this" refer to? If a list of potential participants was generated, this should be stated as well as how many people were on this list. Or were the selection criteria used to make a list of 142 people? It's not clear, so this should be reworded. p. 5, line 31: It was brought up in my previous review that the authors mention that the survey was exempt from HRA/HRB approval. However, this is not the same as research ethics approval. In the letter responding to the previous reviews, the authors provided a clear explanation for why their study was exempt from ethics approval. I believe this is what should be added to the text of the manuscript. Also, the acronyms are not defined anywhere and need to be. Assuming HRB refers to the Irish Health Research Board, there is no indication as to why their approval would be needed in the first place. Did they fund this work? If so, why do the authors state that no funding was obtained?
--

	p. 8, line 54. Table 2 is stated here regarding participant details, but these are in Table 1. This may be an issue with how the manuscript was formatted during the submission process, but the Figures are numbered in the text of the manuscript, but the figures themselves do not carry these labels. p. 9, line 30-34. I found this sentence difficult to follow. I think it would benefit from being reworded. p. 10, line 24-25. I think this sentence would benefit from being reworded. What is “pitfalls” referring to? It is not clear what is being referred to as being outlined. p. 10, line 48. The word “weaponised” is used incorrectly here. “They” refers in context to “members”, and it’s not clear how they could be weaponised. Would replacing “weaponised” with “attacked” be better?
--	--

VERSION 2 – AUTHOR RESPONSE

Replies to reviewer II

1. *<Significant typo correction>*

We are grateful to Prof. Dónal O'Mathúna for his diligence in spotting typos we had become blind to! These are now corrected, and another round of proof-reading was also sought prior to submission. Hopefully any errors are obliterated now, though the editorial office might keep an eye for anything that lurked beyond!

2. *“p. 4, line 48: the sentence ends with saying something will be counteracted. In context, this refers to the experiences of communicating medical science. But why should these experiences be counteracted? It’s not clear that they are all negative, and the results find positive and negative. This sentence should be rewritten.”*

We absolutely agree this sentence was ambiguous – we have rewritten this as;

“Accordingly, the aim of this work is to ascertain the typical experiences of those communicating medical science and identify how negative impacts might be counteracted.”

3. *“p. 5, line 23: I think authors mean “participant selection criteria”, and if so, this word should be added. Note also that criteria is plural and therefore should be followed by “were”... p. 5, line 26: What does “From this” refer to? If a list of potential participants was generated, this should be stated as well as how many people were on this list. Or were the selection criteria used to make a list of 142 people? It’s not clear, so this should be reworded.”*

Again absolutely correct, our apologies – we have rephrased for clarity;

“The participant selection criteria were specifically for those communicating medical science both on social media (defined as having over 1000 followers on twitter) and in mainstream channels (defined

as invited appearances on public television, radio, and / or in the form of newspaper articles & invited comment). With this participant selection criteria, 142 individuals world-wide (from across Europe, America, Africa, and Asia) working predominantly in the English language were identified and invited to partake, of whom 101 (71.1%) responded.”

4. *“p. 5, line 31: It was brought up in my previous review that the authors mention that the survey was exempt from HRA/HRB approval. However, this is not the same as research ethics approval. In the letter responding to the previous reviews, the authors provided a clear explanation for why their study was exempt from ethics approval. I believe this is what should be added to the text of the manuscript. Also, the acronyms are not defined anywhere and need to be. Assuming HRB refers to the Irish Health Research Board, there is no indication as to why their approval would be needed in the first place. Did they fund this work? If so, why do the authors state that no funding was obtained?”*

This is a very good point, and we were sloppy in how we defined this. We have updated our answers to reflect what we clarified in the reply to the reviewer in previous revision. It now reads:

“This survey was undertaken for indicative purposes and was collected from a non-randomised group with no expectation of transferable findings. Accordingly, the Health Research Authority (HRA) decision tool (online at <http://www.hra-decisiontools.org.uk/research/>) indicated specific ethical approval was not required, with the research governance body of Queen's University Belfast (the lead author's primary affiliation at the time) confirming ethical approval for the survey was not required. In all cases, informed consent was sought and obtained prior to subjects partaking, with all data appropriately anonymised. Subject details are given in table 1. “

5. *“p. 8, line 54. Table 2 is stated here regarding participant details, but these are in Table 1.”*

Thank you for spotting this, now corrected.

6. *“p. 9, line 30-34. I found this sentence difficult to follow. I think it would benefit from being reworded.”*

On reflection, we agree entirely. It now reads as a list, with semi-colon breaks for clarity;

“Free-form responses to this question included; improving the media's scientific literacy (false balance and the platforming of anti-science views were repeatedly mentioned); the establishment of legal defence funds; better coordination of professional bodies; robust infrastructure on social media to report disinformation, and better support from police organisations.”

7. *“p. 10, line 24-25. I think this sentence would benefit from being reworded. What is “pitfalls” referring to? It is not clear what is being referred to as being outlined.”*

Again, we agree – revised to:

“It is also important to note that despite the sometimes fraught nature of medical science outreach, a majority of respondents (68.2%, N = 69) found the undertakings rewarding or very rewarding. This is encouraging, but it is crucial we are aware too of the adverse effects that can arise from communicating medical science, many of which are outlined in this manuscript.”

8. *“p. 10, line 48. The word “weaponised” is used incorrectly here. “They” refers in context to “members”, and it's not clear how they could be weaponised. Would replacing weaponised” with “attacked” be better?”*

We were again careless with phrasing – this now reads:

“But rather than be reactionary, it is imperative that bodies and institutions have robust and considered policies for dealing with issues that might arise. The benefits of this are two-fold; firstly, that errant behaviour by members can be corrected. But equally importantly, cognisance of the reality of vexatious complaints also means that bodies and institutions can implement measures to ensure that their procedures cannot be weaponised by malicious actors.”

VERSION 3 – REVIEW

REVIEWER	Dónal O’Mathúna Dublin City University, Ireland and The Ohio State University, USA I know one of the authors from when we worked at the same institution.
REVIEW RETURNED	23-Apr-2020
GENERAL COMMENTS	Thank you to the authors for responding to our earlier review comments and for providing an informative and interesting article.